# Chitosan as a Coating for Biocontrol in Postharvest Products: A Bibliometric Review

**DOI:** 10.3390/membranes11060421

**Published:** 2021-05-31

**Authors:** Ma de la Paz Salgado-Cruz, Julia Salgado-Cruz, Alitzel Belem García-Hernández, Georgina Calderón-Domínguez, Hortensia Gómez-Viquez, Rubén Oliver-Espinoza, María Carmen Fernández-Martínez, Jorge Yáñez-Fernández

**Affiliations:** 1Escuela Nacional de Ciencias Biológicas, Instituto Politécnico Nacional, Ciudad de México 07738, Mexico; mdlpsalgadocr@conacyt.mx (M.d.l.P.S.-C.); agarciah1805@alumno.ipn.mx (A.B.G.-H.); gcalderon@ipn.mx (G.C.-D.); 2Consejo Nacional de Ciencia y Tecnología (CONACYT), Ciudad de México 03940, Mexico; 3Centro de Investigaciones Económicas, Administrativas y Sociales, Instituto Politécnico Nacional, Ciudad de México 11360, Mexico; jsalgado@ipn.mx (J.S.-C.); hgomezv@ipn.mx (H.G.-V.); roliver@ipn.mx (R.O.-E.); 4Laboratorio de Biotecnología Alimentaria, Unidad Profesional Interdisciplinaria de Biotecnología, Instituto Politécnico Nacional, Ciudad de México 07340, Mexico; mfernandezm@ipn.mx

**Keywords:** chitosan, coating, biocontrol, postharvest, fruit, vegetables

## Abstract

The aim of this work was to carry out a systematic literature review focused on the scientific production, trends, and characteristics of a knowledge domain of high worldwide importance, namely, the use of chitosan as a coating for postharvest disease biocontrol in fruits and vegetables, which are generated mainly by fungi and bacteria such as *Aspergillus niger*, *Rhizopus stolonifera*, and *Botrytis cinerea*. For this, the analysis of 875 published documents in the Scopus database was performed for the years 2011 to 2021. The information of the keywords’ co-occurrence was visualized and studied using the free access VOSviewer software to show the trend of the topic in general. The study showed a research increase of the chitosan and nanoparticle chitosan coating applications to diminish the postharvest damage by microorganisms (fungi and bacteria), as well as the improvement of the shelf life and quality of the products.

## 1. Introduction

The most effective and used edible coatings for the protection of fruits and vegetables are made up one or more natural polymers such as cellulose [1], alginate [2], gellan [3], pectin, starch and its derivatives [4], methylcellulose, carboxymethylcellulose [5], Arabic gum [6], whey protein concentrate [7], and chitosan or chitosan nanoemulsion [8,9]. Chitosan is a deacetylated form of chitin, (poly β-(1→4) N-acetyl-d-glucosamine), and is the second most abundant biopolymer found in nature after cellulose, with prominent film-forming properties, non-toxicity, biodegradable and biocompatible properties, high mechanical strength, and excellent antimicrobial activity [10], and it has been used as a coating in various foods [11]. Furthermore, chitosan has been approved by the United States Food and Drug Administration (USFDA) as a food additive and listed as generally recognized as safe (GRAS) in the USA and Japan [6,10].

In the last years, chitosan has gained more attention from researchers due to its broad-spectrum activity and high destruction rate against Gram-positive and Gram-negative bacteria [12] and filamentous fungi [13,14,15]. However, its quality, chemical and biological properties, and therefore its applications are closely related to numerous intrinsic and extrinsic factors such as the degree of deacetylation (DD) [16,17], molecular weight, viscosity, sources, and extraction pathway [18]. Chitosan is a polysaccharide mainly obtained from invertebrates; insect cuticles; fungal cell walls; green algae; yeast; crustacean shells such as those of cockles, shrimp, crabs, etc.; by chemical (alkaline hydrolysis with NaOH solutions) and other less-used methods such as alkaline treatment at high temperature and high pressure [19]; or a process that uses ultrasound [20,21,22] and sometimes, less frequently, enzymatic deacetylation [23]. Moreover, it is important to mention that the method used increases or decreases the deacetylation degree (DD) and determines the content of free amino groups and the cationic character [24].

On the other hand, the DD grade of chitosan indicates the number of acetyl groups removed from chitin, which corresponds to the release of the amino groups from the N-acetylglucosamine monomers. In this regard, He et al. [25] mentioned the chitosan classification according to its DD as being low between 55 and 70%, medium between 70 and 85%, high between 85 and 95%, and ultra-high between 95 and 100% [25,26]. In this case, Kong et al. [27] found that the chitosan microspheres with a 63.6% degree of deacetylation (DD) exhibited the highest antibacterial activity concerning the microspheres made with chitosan at 83.7% of DD, which exerted at least some antibacterial activity.

Furthermore, although the antifungal or antimicrobial mechanism of action of chitosan is still being studied, some hypotheses have already been proposed, e.g., Yang [28] mentioned that this polymer permeates and perforates the fungus nuclei, the protein cell membranes, and intracellular constituents, inactivating bacterial metabolism due to the presence of an amino group that has a positive charge with a pH lower than 6.3 and interacts with negative charges of the cell wall of microorganisms, generating the rupture or lysis of these structures, causing the loss of protein compounds and other intracellular constituents [29]. However, because there are still many factors that need to be analyzed during the coating process such as the coating suspensions properties, the fruit surface microstructure and wettability [1], and the synergy between biocontrol agents and natural bioactive compounds, it is necessary that more studies be conducted [30].

On the other hand, due to the vast amount of existing information, an “intoxication” problem can be generated, as reported by Flórez-Martínez et al. (2021) [31]. Consequently, in this paper, we carried out a bibliometric analysis to show and quantify the evolution of the research, perspectives, challenges, and prospects, not only a cross-sectional study, as this provides limited information.

## 2. Biobliometric Analysis

### 2.1. Steps of Bibliometric Analysis

#### 2.1.1. Methodology of Data Collection

Data for this research were collected from the Scopus database, specifically on 14 April 2021, covering 10 years from 2011 to 2021. For this, a Boolean search string was used. First, the search was realized for those coming from the relevant keyword fields (e.g., chitosan, postharvest, biocontrol, fungi, and phytopathogens). Subsequently, the Scopus service was used, with the option to combine searches, using the “Combine queries” field, where the syntax applied is the # symbol with the “OR” and “AND” operators. The Scopus database query was as follows: ((TITLE-ABS-KEY (chitosan)) AND (TITLE-ABS-KEY (coatings))) AND ((TITLE-ABS-KEY (postharvest OR post-harvest)) OR (TITLE-ABS-KEY (fungal AND diseases)) OR (TITLE-ABS-KEY (biocontrol)) OR (TITLE-ABS-KEY (phytopathogens)) OR (TITLE-ABS-KEY (fungi)) OR (TITLE-ABS-KEY (mould)) OR (TITLE-ABS-KEY (bacterial))) AND (LIMIT-TO (PUBYEAR, 2021) OR LIMIT-TO (PUBYEAR, 2020) OR LIMIT-TO (PUBYEAR, 2019) OR LIMIT-TO (PUBYEAR, 2018) OR LIMIT-TO (PUBYEAR, 2017) OR LIMIT-TO (PUBYEAR, 2016) OR LIMIT-TO (PUBYEAR, 2015) OR LIMIT-TO (PUBYEAR, 2014) OR LIMIT-TO (PUBYEAR, 2013) OR LIMIT-TO (PUBYEAR, 2012) OR LIMIT-TO (PUBYEAR, 2011)) AND (LIMIT-TO (DOCTYPE, “ar”)) AND (EXCLUDE (EXACTKEYWORD, “Human”)) AND (EXCLUDE (EXACTKEYWORD, “Animal”)) AND (LIMIT-TO (LANGUAGE, “English”)) AND (EXCLUDE (EXACTKEYWORD, “Titanium”)) AND (EXCLUDE (EXACTKEYWORD, “Biomedical Applications”)); in order to limit the topic additional phrases, (AND (LIMIT-TO (EXACTKEYWORD, “Fruits”))) was added into the query string, which resulted in 92 articles [32,33].

#### 2.1.2. Methodology of Analysis, Identification and Obtaining Map

The total number of searches resulted in 875 publications (from January 2011 to 14 April 2021). The raw data obtained (CSV Format) from Scopus was analyzed using the VOSviewer software (www.vosviewer.com, accessed on 25 March 2021; Van Eck and Waltman, 2009–2020, version 1.6.15, Leiden University, Leiden, The Netherlands) for the construction and bibliometric visualization of networks of institutions, countries, keywords, and citations per article.

#### 2.1.3. Methodology of Analysis of Further Analysis

SigmaPlot^®^ software (SigmaPlot 12.0, Systat Software, Inc. SigmaPlot for Windows. San Jose, CA, USA) was used to design and produce the graph shown in Figure 1; then, the “Dynamic fit wizard” plugin was applied for curve fitting, using a linear regression model. The obtained equation was Equation (1).
(1)$$f=y_{0}+a*x$$
where *Y* = response (number of published papers); *x* = years; *a* = the line’s slope, a parameter that describes the steepness of the curve; and *y*_0_ = intercept.

## 3. Results

### 3.1. Scientific Production Period

Figure 1 shows the number of scientific publications per year, limiting this research to a period from January 2011 to 14 April 2021. It can be observed that the temporal evolution between the number of articles versus years, in this field of research, has had linear growth in the last decade. This data indicates that researchers have focused on increasing the number of publications on this topic due to the use of chitosan coatings as a preservation technique potentially decreasing antimicrobial and antifungal activity when applied to different fruits and vegetables. From 2011 to 2013, few documents per year were published on the subject (11, 34, and 41, respectively), but these have increased. It should be noted that the year 2021 only covered the period until April. In order to perform the data analysis and have it be explained by a linear regression model using the best fit (*R*^2^ = 0.9859) of the results, we did not consider this year and only included studies until the year 2020, as shown in Figure 1. According to the equation obtained for the year 2021, 158 or 159 published documents are forecast. However, it is an issue that requires intensive research, and therefore a greater number of publications on the topic is expected.

### 3.2. Keyword Analysis

In all published documents, the central focus of an article is highlighted using keywords, which are essential and facilitate mapping for readers [32] so that their analysis is necessary. One method is by word cloud (Figure 2), which provides a first view of the dataset, allowing us to explore and visually analyze, as well as to size and create the first classification for our data. The size of the words “chitosan” and “coatings” suggests that in most research, these words have been the most persistent theme. However, this technique only provides qualitative information, and therefore it is necessary to execute a more in-depth analysis.

A more accurate method is the co-keyword cluster mapping (Figure 3), obtained from author keywords. Here, the software (VOSviewer) analyzed the 84 most frequently terms, and each one of them was repeated at least six times. It is imperative to mention that each circle represents a keyword, and its size indicates its appearance frequency in the articles. Data analysis generated nine clusters marked with different colors, e.g., the first cluster (in red) contained 19 terms, with “chitosan” being the most frequent term, with a greater node keyword, closely associated with the 81 terms belonging at the nine clusters, but mainly with largest nodes such as “edible coating, coating, shelf life, antibacterial activity”.

VOSViewer software can also reflect the trend, impact and evolutionary process of the topic’s high-frequency keywords involving “chitosan” and its many applications. The overlay display map, showing the gradient color from blue to yellow, indicated the average citation score of a keyword reported by Guo et al. [34] (Figure 4). It is important to note that the node’s color also determined when the term or keyword was introduced for the first time in the network [14]. Our analysis allowed us to visualize the fact that these issues will continue to take hold in the future. In particular, it was observed that from 2019, the topics “shelf life, preservation, nanoparticles” and “antibacterial properties” began to gain greater importance. These results could be significant for the scientific development research that involves the topic of food waste that generates so many economic losses. On the other hand, the most important group of terms is related to studies where the co-occurrence network is present for words such as “edible coating”, “antifungal”, “postharvest”, and “antibacterial or antimicrobial activity”. These words seem to be an issue related to increasing the shelf life of fruits and vegetables, as shown in the literature review carried out in Section 3.4. The most important terms in the keyword map can be an idea generator for researchers. For example, the generation of coatings involved added nanoparticles for protection and longer shelf life of different foods. However, much research is still needed to establish the existing interactions between food matrices and these coatings.

On the other hand, as shown in Figure 4, the keyword “chitosan”, highlighted with the larger circle in blue, also determined a central position, indicating its importance and direct connection with other smaller nodes such as “fruit coating”, “useful life”, “strawberry”, “mango”, “guava”, “tomato”, and “papaya”. Terms that gain importance as will be seen later.

### 3.3. Keyword the Top 20 Most-Cited Documents

On the other hand, Table 1 shows the top 20 most-cited articles, extracted from the search of 875 documents. Obtained data such as the year of publication, authorship, journal title, publication count, and citation count were analyzed. Due to the high quantity of published papers, our analysis focused on the most highly cited papers and those related to the keyword “fruits”, which generated 92 documents, analyzed as described in Table 2, Table 3 and Table 4.

**Table 1 membranes-11-00421-t001:** The top of 20 most cited authors and documents between 2011 and 2021.

Document Title/Journal	Total Citations	Cite Score 2019	Journal’s Impact Factor	Reference
Antimicrobial activity of iron oxide nanoparticle upon modulation of nanoparticle-bacteria interface/*Scientific Reports*	258	7.2	4.576	[35]
Oxidative stress induced by inorganic nanoparticles in bacteria and aquatic microalgae—State of the art and knowledge gaps/*Nanotoxicology*	202	11.5	4.925	[36]
Development of noncytotoxic chitosan-gold nanocomposites as efficient antibacterial materials/*ACS Applied Materials and Interfaces*	152	13.6	8.758	[37]
Antimicrobial Electrospun Biopolymer Nanofiber Mats Functionalized with Graphene Oxide-Silver Nanocomposites/*ACS Applied Materials and Interfaces*	146	13.6	8.758	[38]
Chitosan and chitosan-ZnO-based complex nanoparticles: Formation, characterization, and antibacterial activity/*Journal of Materials Chemistry B*	125	8.8	5.344	[39]
Effect of chitosan coatings on the physicochemical characteristics of Eksotika II papaya (*Carica papaya* L.) fruit during cold storage/*Food Chemistry*	218	10.7	6.306	[40]
Effect of hydroxypropylmethylcellulose and chitosan coatings with and without bergamot essential oil on quality and safety of cold-stored grapes/*Postharvest Biology and Technology*	196	7.8	4.303	[41]
Advanced physico-chemical characterization of chitosan by means of TGA coupled on-line with FTIR and GCMS: Thermal degradation and water adsorption capacity/*Polymer Degradation and Stability*	192	6.8	4.032	[42]
Development of edible bioactive coating based on modified chitosan for increasing the shelf life of strawberries/*Food Research International*	166	6.2	4.972	[43]
Effects of chitosan coating on postharvest life and quality of guava (*Psidium guajava* L.) fruit during cold storage/*Scientia Horticulturae*	162	3.7	2.769	[44]
Production and evaluation of dry alginate-chitosan microcapsules as an enteric delivery vehicle for probiotic bacteria/*Biomacromolecules*	158	10	6.092	[45]
Effect of chitosan edible coating on the quality of double filleted Indian oil sardine (*Sardinella longiceps*) during chilled storage/*Food Hydrocolloids*	155	10.6	7.053	[46]
Antimicrobial edible films and coatings for fresh and minimally processed fruits and vegetables: A review/*Critical Reviews in Food Science and Nutrition*	154	7.862	13.2	[47]
Effect of chitosan-based edible coating on antioxidants, antioxidant enzyme system, and postharvest fruit quality of strawberries (*Fragaria × aranassa* Duch.)/*LWT—Food Science and Technology*	152	6.4	4.006	[48]
Antimicrobial activity of chitosan, organic acids and nano-sized solubilisates for potential use in smart antimicrobially-active packaging for potential food applications/*Food Control*	146	8.4	4.258	[49]
Comparison of chitosan-gelatin composite and bilayer coating and film effect on the quality of refrigerated rainbow trout/*Food Chemistry*	120	10.7	6.306	[50]
Antimicrobial effectiveness of bioactive packaging materials from edible chitosan and casein polymers: Assessment on carrot, cheese, and salami/*Journal of Food Science*	118	3.7	2.478	[51]
Effect of chitosan-aloe vera coating on postharvest quality of blueberry (*Vaccinium corymbosum*) fruit/*Postharvest Biology and Technology*	117	7.8	4.303	[52]
Survivability of probiotics encapsulated in alginate gel microbeads using a novel impinging aerosols method/*International Journal of Food Microbiology*	113	7.4	4.187	[53]
Effects of carboxymethyl cellulose and chitosan bilayer edible coating on postharvest quality of citrus fruit/*Postharvest Biology and Technology*	109	7.8	4.303	[54]

It should be noted that the 20 most-cited articles were all published between 2011 and 2015, with a citation range from 109 to 258, where the highest rated was Arakha et al. (2015) [35]. This study was published in *Scientific Reports* and intended to explore the interaction pattern role of the iron oxide nanoparticle (IONP)–bacteria interface that enhances the antimicrobial activity of IONP using positively charged chitosan. In analyzing the rest of the authors and the most cited scientific papers in the domain under study, we noted the importance of the use of chitosan and its multiple applications as well as their effect as a coating in various fruits. In this sense, the second most cited document [40] reported the use of chitosan as an effective control in reducing weight loss, maintaining firmness, delayed changes in the peel color, and soluble solids in papaya (*Carica papaya* L.), which is one of the most important fruit crops in the world and has a short post-harvest life. However, it did not study the damage by opportunistic plant pathogens capable of producing diseases or loss of crops, which has led to countless studies, as shown in Table 2, Table 3 and Table 4.

It is worth noting that the emerging interdisciplinary field of nanotechnology has been a recurring phenomenon in recent studies, as shown in Table 1, with the highest cited document or the documents published by various authors [36,37,38,39].

### 3.4. Review of Documents with Keyword “Fruits”

Exceptionally, the co-occurrence between keywords allows for the generation of knowledge in search of a common goal. Over the past decade, researchers around the world have developed many different methods to minimize postharvest fruit loss because they have the highest waste rates of any food product (45% waste [55]), which is a global problem. A novel method is the use of chitosan coatings as well as their different combinations with other polymers or with essential oils or nanoparticles, among others, as shown in Table 2. This allows for the storage period to be increased in order to postpone the deterioration of fruits and vegetables and preventing the growth of microorganisms transmitted by food on the surfaces of the products.

**Table 2 membranes-11-00421-t002:** Fungi that affect postharvest fruit quality: analysis from 2011 to 2021.

Fungi	Disease or Damage	Fruit	Coatings	Reference
*Aspergillus niger* ^A^	Gray mold	Strawberry (*Fragaria ananassa*)	Chitosan incorporated with olive oil residues	[56]
*Rhizopus stolonifera* ^B^	Brown spots and softening by rotting		Chitosan as gel, nanoscale particles or nanocomposite	[13]
*Botrytis cinerea* ^C^	Black mold (black rot)		Coatings with cellulose, chitin, and chitosan nanomaterials	[1]
			Chitosan functionalized by acylation with palmitoyl chloride and essential oils of limonene and peppermint	[43]
		Blueberries and cherry tomatoes	Chitosan thymol nanoparticles prepared by ionic gelation	[57]
		Cherry tomatoes	Thymol nanoemulsions incorporated in quinoa protein/chitosan edible films	[58]
*P. expansum*	Blue mold	Apples (*Malus domestica* Borkh. cv. Gala)	Heating at 38 °C and 1% chitosan	[59]
			Chitosan (medium molecular weight with 60% or more deacetylated)	[60]
*P. citrinum*		Lingwu long jujube fruit	Chitosan and cinnamon oil	[61]
*Alternaria alternate*	Black mold	Pitaya (*Stenocereus griseus* H.)	Chitosan + oleic acid	[62]
		Bell pepper (*Capsicumannuum* L.)	Chitosan nanoparticles with α-pinene	[63]
*Colletotrichum gloeosporioides*	Anthracnose	Guava (*Psidium guajava* L.)	Chitosan–citric acid	[64]
		Papaya (*Carica papaya* L.)	Chitosan and *Mentha villosa* Huds or *M. piperita* L. essential oil	[65,66]
		Mango (*Mangifera indica* L.)	Chitosan with thyme oil	[67]
			Vanillin-chitosan and zeolite or activated carbon	[68]
			Chitosan, carboxymethyl cellulose, and vanillin	[69]
		Avocado (*Persea americana*)	Chitosan nanoparticles and chitosan biocomposites with pepper tree essential oil	[70]
		Papaya (*Carica papaya* L.)	Aloe vera–chitosan composite	[71]
*Colletotrichum fragariae*	Anthracnose crown rot	Strawberry (*Fragaria ananassa* Duch)	Chitosan functionalized with cinnamon essential oil and aqueous extract of Roselle calyces	[72]
*Aspergillus flavus*	Production of aflatoxins	Fig fruit	Chitosan and propolis nanoparticles	[8]
*Fusarium solani*	Lesions on roots	Cucumber (*Cucumis sativus* L.)	Nanostructured chitosan and chitosan functionalized with cinnamon essential oil or trans-cinnamaldehyde	[15]
*Fusarium oxysporum*	Wilt	Watermelon (*Citrullus lanatus*)	Chitosan-mesoporous silica nanoparticle	[73]
*Burkholderia seminalis*	Fruit rot	*Apricot fruit*	Acid-soluble and water-soluble chitosan	[74]

The letters correspond to the fungi worked by each author: Letter A, B, and C correspond to [13]; letter B corresponds to [56], and letter C corresponds to [1].

It was observed that there is plenty of research involving published studies concerning the antimicrobial and antifungal activity of chitosan as well as a combination with other polymers or the application of different essential oils against foodborne pathogens. However, of the total reports (875 documents published), only 93 documents with the keyword “fruits” were analyzed due to the importance of this kind of food. The findings mentioned below and those in Table 2, Table 3, Table 4 and Table 5 correspond to these documents.

**Table 3 membranes-11-00421-t003:** Published results of bacterial contamination by different microorganisms in fruits: from 2011 to 2021.

Bacteria	Fruit	Coatings	Reference
*Staphylococcus aureus*, *Escherichia coli*, and *Bacillus subtilis*.	Snake fruit, *Salacca zalacca*	Glucomannan–beeswax–chitosan	[75]
	Bananas (*Musa acuminata* L.)	ZnO nanoparticles incorporated into chitosan/Arabic gum	[6]
*Staphylococcus aureus*, *Listeria monocytogenes*, *Pseudomonas aeruginosa*, *Salmonella* spp., *Escherichia coli*	Grapes	Chitosan nanoparticles	[76]
*E. coli*, *S. aureus*, *B. subtilis*, and *M. guilliermond**ii*.	Mango (*Mangifera indica* L.)	Ferulic acid-grafted chitosan using recombinant bacterial laccase from Bacillus vallismortis	[28]
*Salmonella typhimurium*, total mesophilic aerobes, yeasts, and molds	Grape berries (*Vitis vinifera* L. × *V. labruscana* Bailey)	Lemongrass oil–chitosan emulsion	[77]
*Staphylococcus aureus*, *Escherichia coli*, *Listeria innocua*	Watermelon, melon, strawberries	Nanoparticles of vanillin are formed in situ from an aqueous/ethane solution and deposited on the surface of chitosan, using a high-intensity ultrasonic method	[78]
*Staphylococcus aureus*, *Escherichia coli*	Bananas	Carboxymethyl cellulose on quaternized chitosan (2-N-hydroxypropyl-3-trimethylammonium chloride chitosan, HTCC)	[79]
*Bacillus cereus*, *B. subtilis*, and *Serratia marcescens*	Mangaba fruits	Cassava starch, chitosan, and Myrcia ovata Cambessedes essential oils	[80]
*Escherichia coli* O157:H7	Cherry tomato	Chitosan with Artemisia annua oil	[81]

**Table 4 membranes-11-00421-t004:** Published results from bacterial and fungal contamination by different microorganisms in fruits: from 2011 to 2021.

Psychrophilic Bacterial, Mesophilic Aerobic, Yeast, and Mold		
Apricot fruits (*Prunus armeniaca* L. cultivar Rival)	Chitosan enriched with pomegranate peel extract	[9]
Blueberry fruit (*Vacciniumashei* L.)	Chitosan with nano-material films such as silicon and titanium dioxides	[82]
Blueberry (*Vaccinium corymbosum*)	Chitosan/nano-titanium dioxide and chitosan/nano-titanium dioxide (tween-thymol)	[83]
Black mulberry (*Morus nigra*)	Chitosan and cassava starch	[84]
Tomato (*Solanum lycopersicum* L.)	Chitosan–Ruta graveolens essential oil coatings	[85]
Cucumber (*Cucumis sativus* L.)	Nanoparticles and Zataria multiflora essential oil	[86]
Strawberries (*Fragaria ananassa* cv. Camarosa)	Natamycin, nisin, pomegranate, and grape seed extract in chitosan	[87]
Strawberries	Chitosan-monomethyl fumaric acid	[88]
Fresh-cut apple slices	Chitosan and stevia	[89]
	110 and 300 nm chitosan nanoparticles or chitosan dissolved in 2% citric acid	[90]
Fig (*Ficus carica* L.)	Chitosan, thymol, and their combination	[91]
Tomatoes (*Lycopersicon esculentum* Mill.)	Cassava starch–chitosan enriched with Lippia sidoides Cham. essential oil and pomegranate peel extract	[92]
Kiwifruits (*Actinidia deliciosa* cv. Hayward)	Aloe vera, chitosan (formulated with acetic or citric acid), and sodium alginate	[93]
Guava (*Psidium guajava* L.)	Chitosan–cassava starch coatings containing a mixture of Lippia gracilis Schauer genotypes	[94]
Wolfberry (*Lycium barbarum* L. cv. Ningqi No. 1)	Hot water dip at 42 °C for 30 min and 1% chitosan	[95]
**Molds and Yeasts**		
Tomato (*Lycopersicon esculentum*)	Chitosan b enriched with pequi peel extract	[96]
Strawberries (*Fragaria* × *ananassa*)	Peony extracts (*Paeonia rockii*) dispersed in chitosan	[97]
	Quinoa protein–chitosan–sunflower oil	[97]

Recently, the impact of preharvest foliar spraying with chitosan and postharvest aloe vera gel coating (AVG) on the quality of table grapes during storage was evaluated, thereby extending the shelf life of the fruit up to 15 days by significantly reducing the decomposition index [99]. The relevance of this study marks an important stage in the supply chain (pre-harvest) in which there is little research. Another recent finding is the production of edible coating films based on Pickering emulsions, which showed a smaller droplet size, narrower size distribution, and improved stability. These could inhibit the growth of typical spoilage organisms such as S. aureus and E. coli in order to preserve fruits and vegetables [100]. In 2020, Jung et al. [101] applied this method by adding oleic acid and cellulose nanocrystal in “Bartlett” pears (*Pyrus communis* L.) for delaying ripening and superficial scald during the long-term cold storage.

Table 2, Table 3 and Table 4 show studies concerning the application of chitosan as an antimicrobial and antifungal to maintain fruit and vegetable quality at the postharvest stage. It is highlighted that several studies have focused on reducing the antifungal activity of *Aspergillus niger*, *Rhizopus stolonifera*, *Botrytis cinerea*, *P. expansum*, *Alternaria alternate*, *Colletotrichum gloeosporioides*, etc. In some studies, the antifungal activity of chitosan depends on the extraction procedure, the deacetylation percentage, molecular weight, or the microstructure of the fruit and the interaction of the coating material.

Some fruits such as strawberries [1,13,43,48,56,72,87,98,102,103,104], mango (*Mangifera indica* L.) [28,68,105,106,107,108,109], tomato [58,81,85,92,96,110,111,112,113], guava [64,94,114], banana [6,79,108], apples [59,60,115,116], or fresh-cut apple slices [89,90] are especially perishable and therefore there is a larger number of documents focused on decreasing mechanical injury, desiccation, decay, and physiological disorders during storage [43], as observed in Table 2, Table 3, Table 4 and Table 5. In this sense, the most cited document (166 citations) [43] mentioned that the use of chitosan functionalized by acylation with palmitoyl chloride increase its hydrophobicity in order to ensure a controlled release and improve its stability and adherence in strawberries. Notwithstanding, research in this area is still incipient, and therefore is necessary to carry out future research about the topic with other fruits or in different matrixes of food.

For this purpose, mixtures of chitosan and some other materials have also been used, as shown in Table 5; these results indicated that the coatings could reduce the damage in different fruits or vegetables.

Moreover, other studies have addressed that the chitosan coating applied in pummelo fruit mitigates the development of juice sac granulation and delays postharvest senescence in the same fruit during room temperature storage [117], and in eggplant cultivars (purple long, purple round, and white long), chitosan was effective in minimizing weight loss, maintained quality, and prolonged storability with good appearance and overall acceptability [118]. However, it is necessary to conduct more research focused on combinations of adequate techniques and different coating materials that consider the intrinsic and extrinsic factors that affect food, as well as allowing for enhancement of shelf life and decreases in the amount of waste.

**Table 5 membranes-11-00421-t005:** Coating materials mixture with chitosan applied to extend the shelf-life and improve the quality of fruits.

Fruit	Coatings	Results	Reference
Le Conte pears	Chitosan–beeswax-based	The use of coatings improved quality parameters by successfully showing a decrease in weight loss, deterioration, and softening rate.	[119]
Strawberries	Chitosan and apple peel polyphenols composite	The weight loss, decay percentage, and senescence were reduced and maintained quality attributes of the fruits during storage.	[120]
	Chitosan–whey protein isolate	A considerable reduction in color indices, weight loss, pH, and titratable acidity; reduction in sugars, ascorbic acid, and total phenolics was noted.	[102]
	Three different forms of chitosan by decoloration method, without the decoloration step and the deproteinization step	Chitosan coatings delayed changes in weight loss and the appearance of fungal infection.	[103]
Strawberries (*Fragaria* × *ananassas* Duchesne ex Rozier ‘Earliglow’)	Chitosan solutions of 0.5, 1.0, and 1.5 g/100 mL	Coatings can maintain high antioxidant levels and high-antioxidant enzyme activities and inhibit increased oxidative enzyme activity to reduce moisture loss and delay senescence.	[48]
Strawberries (*Fragaria* × *ananassa* cv. Camarosa)	Chitosan–lemon essential oil	Pure chitosan promoted the formation of esters and dimethyl furfural, while coatings containing lemon essential oil incorporated terpenes (limonene, γ-terpinene, p-cymene, and α-citral) to the volatiles of the fruit and improved the fermentation process, modifying the typical fruit aroma composition.	[104]
Mango (*Mangifera indica* L.)	Chitosan–aloe vera gels and calcium chloride (CaCl_2_)	The results showed a decrease in weight loss, reduction of ascorbic acid, and inhibition of polyphenol oxidase (PPO) activity during the storage period.	[105]
	Chitosan–cinnamon essential oil microcapsules	Multilayer coatings made by electrostatic interaction on mangoes slowed down the increase in weight loss and preserved firmness under storage conditions.	[106]
	Chitosan (1, 2, or 3%)	Chitosan delayed the climacteric peak, water loss, firmness, and sugar content, as well as decreasing starch degradation, and it was also observed to affect basic mitochondrial respiration.	[107]
	Chitosan, gallic acid, and chitosan gallate	The coatings delayed ripening and weight loss and maintained a higher peel membrane stability index as well as the quality of the ‘Hindi-Besennara’ mangoes during 2 weeks of shelf life.	[108]
	Chitosan solutions of high, medium, and low molecular weight	The film-forming properties of chitosan were influenced by molecular weight and significantly affected the postharvest quality of mango fruit during storage.	[109]
Apricots	Alginate, chitosan, and gellan gum	The coating prolongs the shelf life and inhibits oxidative enzymes, specifically peroxidase (POD) and polyphenol oxidase (PPO).	[3]
Guava (*Psidium guajava* L.)	Chitosan (1%, 2%, or 3%)	Chitosan suppressed respiratory rate, fresh weight loss, firmness, and skin color with delayed degradation of chlorophyll.	[114]
Tomato (*Solanum lycopersicum* L.)	Chitosan (1.5%)	The coating is effective in maintaining less weight loss, having more firmness and slowing the nutraceutical loss that occurs in the postharvest, mainly of the carotenoid lycopene.	[110]
Cherry tomato	Palm stearin, palm kernel olein (PSPKOo), and chitosan of different degrees of deacetylation (DD) (85 and 95%)	Chitosan film with 85% DD (MW 300,000 Da) and 31% PSPKOo blend was the most effective in reducing weight loss and maintaining firmness and redness.	[111]
Chinese kiwifruit (*Actinidia chinensis* Planch)	Chitosan enriched with salicylic acid	The treatment significantly maintained texture and color, inhibited moisture loss and acidity change, and delayed the decomposition of vitamin C and soluble solids.	[121]
	Chitosan with some olive waste extracts of leaf and pomace extracts	Chitosan coating films significantly reduced the gradual decrease in total phenolics, flavonoids, and antioxidants, and relatively improved the nutritional quality of apple during postharvest.	[115]
Apple (*Malus domestica* var. Anna) Apples (cv. Golab Kohanz)	Nanochitosan emulsion (0.2 and 0.5%)	The effect of nanochitosan coating was shown to meaningfully reduce the weight loss, respiration rate, ethylene production, and peroxidase activity of the samples compared to the control.	[116]
Longan fruit (*Dimocarpus longan*)	UV-C irradiation and carrageenan and chitosan-based coating	The application of UV treatment followed by chitosan coating was the best treatment combination for control enzyme activities and reduced the rate of senescence.	[122]
Pomegranate (*Punica granatum* L.)	Resin wax (Britex Ti), carnauba wax (Xedasol M14), and chitosan (1 and 2% *w*/*v*)	The coated fruits showed significantly lower respiration rate and weight loss, but the carnauba wax was able to maintain considerably higher fruit quality and bioactive compounds.	[123]
Carambola (*Averrhoa carambola* L.)	Chitosan, Arabic gum, and alginate	The coated fruits showed a significant delay in the change of weight loss, percentage of decomposition, accumulation of sugar, degradation of pigments, and content of ascorbic acid, maintaining the highest concentration of total phenols.	[124]
Tomatoes	Ultrasound-assisted chitosan surfactant nanostructure (micelle sizes of 400, 600, and 800 nm)	The treatment enhanced the phenolic content while maintaining a lower respiration level throughout most of the storage duration. However, the weight loss was greater in the treated fruits.	[112]
Grape (*Vitis vinifera* (*V. vinifera*))	Putrescine alone or with chitosan	The chitosan–putrescine combination reduced weight loss, incidence of decay, browning, and berry breakage and cracking.	[125]
	Chitosan (0.5 or 1%)	The treated berries showed less weight loss, decay, browning, shattering, and cracking.	[126]
Longan (*Dimocarpus longan* Lour.)	Chitosan/nano-silica hybrid filmusing tetraethoxysilane as precursor	The film remarkably prolonged shelf life, reduced browning index, delayed weight loss, and inhibited the increase in malondialdehyde amount and polyphenoloxidase activity in fresh fruit.	[127]
Tomato fruit (*Lycopersicon Esculentum*)	Chitosan and a chitosan derivative (N,O-carboxymethyl chitosan)	The coating can extend the shelf life and improve the quality of tomato fruit by delaying ripening, reducing weight loss, and preserving the fruit firmness.	[113]
Yali pears (*Pyrus bretschneideri* Rehd.)	Chitosan (1.5%)	Chitosan treatments both before and after damage delayed the color changes caused by damage, inhibited increase disease incidence, and improved the bruise recovery during the storage.	[128]
Papaya (*Carica papaya* L.)	Chitosan (95% deacetylated; 0.5, 1.0, 1.5, and 2.0% *w*/*v*)	Chitosan provided effective control to reduce weight loss, maintained firmness, and delayed changes in the peel color and soluble solids concentration during 5 weeks of storage.	[40]

## 4. Conclusions

This bibliometric review analyzed the evolutionary process over the past decade of topics about chitosan as coating and their fruits and vegetables’ antifungal or antimicrobial effects. VOSviewer software is a useful and versatile tool that allows for easy visualization and analysis of bibliometric networks. In this paper, 875 documents reported that coatings made of chitosan only or chitosan in combination with other biopolymers are a natural and safe post-harvest biocontrol strategy to decrease microbial spoilage mainly by pre- and post-harvest diseases, reducing the damage of fruits as well as extending their shelf life. Finally, this work can provide a useful perspective for future research in the studied field since it demonstrates the existence of an emerging area of study that is intended to reduce a global problem caused by the generation of agro-industrial waste due to the loss of post-harvest damaged crops.

## Figures and Tables

**Figure 1 membranes-11-00421-f001:**
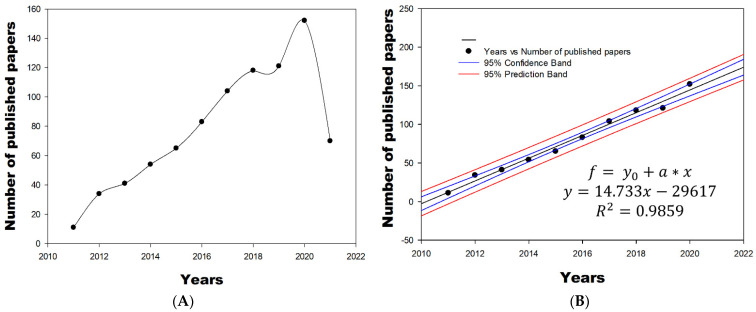
(**A**) Number of articles in Scopus from 2011 to 2021 in the field of the use and applications of postharvest chitosan products. (**B**) Fitted linear trend for the number of publications.

**Figure 2 membranes-11-00421-f002:**
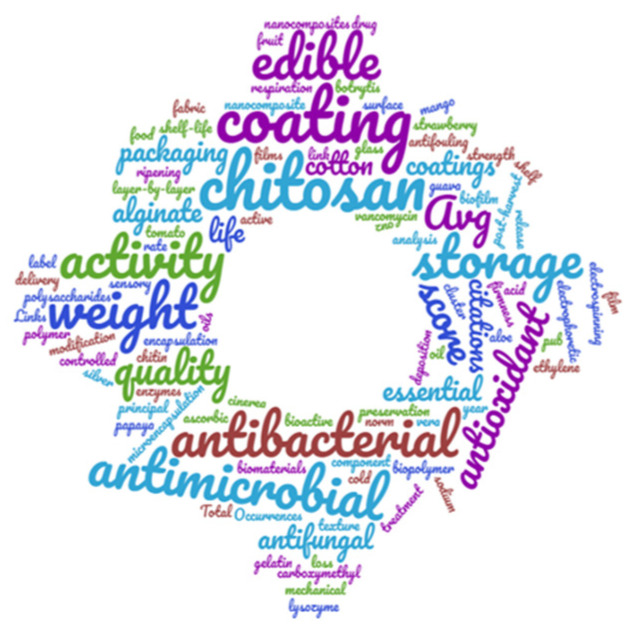
Word cloud extracted from nubedepalabras.es.

**Figure 3 membranes-11-00421-f003:**
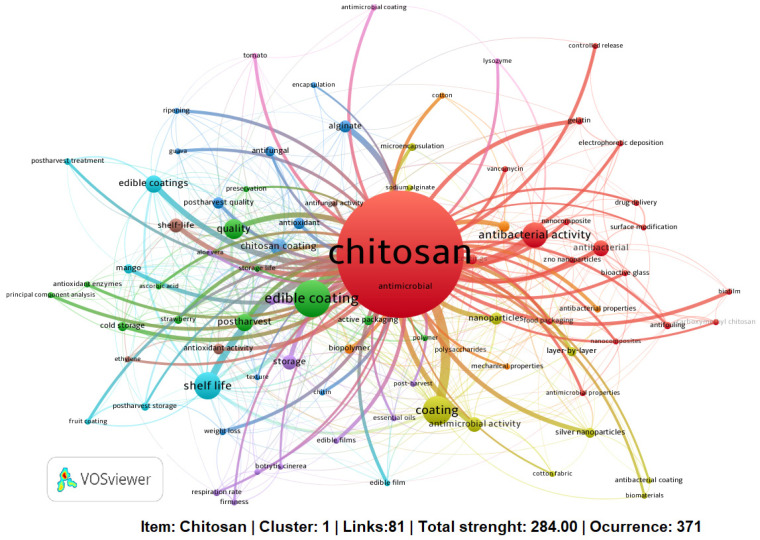
Map of co-occurrence network visualization based on article weights for the terms related with the first group.

**Figure 4 membranes-11-00421-f004:**
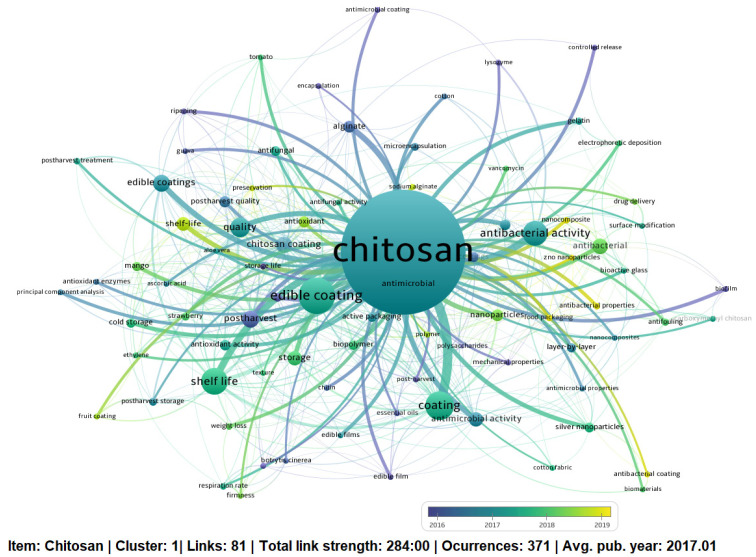
Map of co-overlay visualization of author keywords from 2010 to 2021.

## Data Availability

Not applicable.

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
