# Peer review of "Chitosan as a Coating for Biocontrol in Postharvest Products: A Bibliometric Review"

_membranes, 2021, doi:10.3390/membranes11060421_

Round 1

Reviewer 1 Report

(1) It is recommended that the authors can illustrate some representative works in Tables 1-5. 

(2) The language editing is needed. 

Thanks. 

Author Response

Thanks for your suggestions. We try to answer your questions.

Reviewer 2 Report

This manuscript represents a neat manuscript on the role of chitosan as a post-, and perhaps pre-harvest preserver. It deals with a high number of citations, which is an important aspect of this type of paper.

Although currently thought of as a non-conflicting compound, as far as human health is concerned, I would have liked that the authors would have included a short section on this topic, perhaps at the end of the manuscript. 

I

Author Response

(The authors gave the same response as above.)

Reviewer 3 Report

An interesting summary of the research from the last decade. The article may be a good resource for formulating chitosan-based coatings. However, some sentences should be more precise as they give rise to some ambiguities (writing style).

Title of article - proposing to use “fruits and vegetables” instead of the word “products”

Abstract

Please correct the first sentence "The aim of this word…" (L 15-17) this sentence should be rewritten (writing style).

L 20-22. What about shelf life and quality of these kinds of products? It should be mentioned about it (Table 5).

Keywords - Add words: fruit, vegetable

Introduction

L 30. Is better to use “chitosan” than “the latter”

L 31-32. Which biopolymer is the first?

M&M

L 78. 'was' instead of 'is'

Results

L 128. According to the equation presented in Figure 1 it should be 158 or 159 of the publication. Please check.

L 264. Vegetable? Authors use the keyword “fruits” (L 249). Please explain it in the paragraph. The same is concern for Table 2, 3, 4, 5. Why Authors use word "vegetables" in the title and inside the tables - name of column?

Table 5. The title is poorly formulated. The tables 2, 3, 4 also present the coatings materials. It should be improved. e.g. Coatings materials mixture with chitosan applied to extend the shelf life and improve the quality of fruits.

Table 2, 3, 4, 5. The word “chitosan” is used replaceable in the tables (the coatings column) with its abbreviation (Ch), what does it depend on? Standardize.

Table 5. Please check reference 106 page 11 - coatings - whey protein isolate? WPI was used alone or with chitosan?

L 333-339. Text is duplicated in the conclusion. Remove.

Conclusions

Please add the conclusions based on Table 5.

Other comments:

Please remove dot in the title of article and in the title of section 2.1.1. and 2.1.2.

Author Response

(The authors gave the same response as above.)

Reviewer 4 Report

The manuscript entitled "Chitosan as a coating for biocontrol in a postharvest products: a bibliometric review" is a valuable contribution to the practically relevant and scientifically important topic of biomaterial and biotechnologies development. The paper is written as a bibliometric review. It provides an insight into  the current progress in chitosan's practical applications and more specifically into it's use for coatings of vegetables and fruits. Generallly, the manuscript is coherent and nicely written, so I don't see major obstacles to  prevent it's publishing in the "Membranes". However, a number of issues should be adressed before it can be accepted for publication.

The balance (length) of each part of the manuscript  (Abstract, Intro e.t.c) should be double checked. As, for example, the abstract part is very short.

Then the accuracy of the English language should be checked by a native speaker.

Finally, there are so many table in the manuscript, and I suggest to moove some of those ro supplements if the authors think that it is apropriate.

The other minor comments:

Line 3-4: In my opinion, the authors’ name should be given in full.

line 15: not “word” but revision or study or work

The abstract part is too short, it should be increased.

line 30: remove “among others.”

line 34: start the new sentence after [10]

line 38: no need for comma

lines 41-46: the language is too long, should be split to several smaller ones

line 67: comma before and

line 70: a year of the publication should be given

line 78: 14th

line 181: figure, the words are too small and not all the words are nicely visible

line 212: same comment

line 222: the highest rated paper?

line 233: I suggest substituting the “authors” for the “reference” in the table; and write please “journal’s impact factor.”

line 236: what does the word “outstandingly” mean? I suggest substituting it

line 242: “to stop and delay” =to postpone

line 245: remove “exists.”

line 264: at the postharvest stage

line 288: it is necessary to conduct more research.

line 289: “contemplate” is a hard word, please substitute it

line 347-348: the last sentence can be stronger.

Author Response

(The authors gave the same response as above.)

Reviewer 5 Report

The manuscript "Chitosan as a coating for biocontrol in postharvest products: a bibliometric review " offers a bibliometric analisys of literature about the use ofchitosan as a coating for postharvest diseases biocontrol in fruits and vegetables. Unexpectedly it provides much more information than "traditional" review and with immediate impact. Table 5 is very useful. Only maps obtained using VOSviewer software should be better explained.

Author Response

(The authors gave the same response as above.)

Round 2

Reviewer 4 Report

The manuscript has been improved.